# Application of Infrared Multiple Photon Dissociation (IRMPD) Spectroscopy in Chiral Analysis

**DOI:** 10.3390/molecules25215152

**Published:** 2020-11-05

**Authors:** Yingying Shi, Mengying Du, Juan Ren, Kailing Zhang, Yicheng Xu, Xianglei Kong

**Affiliations:** 1State Key Laboratory of Elemento-Organic Chemistry, College of Chemistry, Nankai University, Tianjin 300071, China; 2120170819@mail.nankai.edu.cn (Y.S.); 2120200778@mail.nankai.edu.cn (M.D.); juanren1227@foxmail.com (J.R.); zhkl@tju.edu.cn (K.Z.); 2120190792@mail.nankai.edu.cn (Y.X.); 2School of Precision Instrument and Opto-Electronics Engineering, Tianjin University, Tianjin 300072, China; 3Collaborative Innovation Center of Chemical Science and Engineering, Nankai University, Tianjin 300071, China

**Keywords:** IRMPD spectroscopy, chirality, mass spectrometry, diastereomeric ions

## Abstract

In recent years, methods based on photodissociation in the gas phase have become powerful means in the field of chiral analysis. Among them, infrared multiple photon dissociation (IRMPD) spectroscopy is a very attractive one, since it can provide valuable spectral and structural information of chiral complexes in addition to chiral discrimination. Experimentally, the method can be fulfilled by the isolation of target diastereomeric ions in an ion trap followed by the irradiation of a tunable IR laser. Chiral analysis is performed by comparing the difference existing in the spectra of enantiomers. Combined with theoretical calculations, their structures can be further understood on the molecular scale. By now, lots of chiral molecules, including amino acids and peptides, have been studied with the method combined with theoretical calculations. This review summarizes the relative experimental results obtained, and discusses the limitation and prospects of the method.

## 1. Introduction

As a basic attribute of nature, chirality is of great significance in the fields of chemistry, medicine, and life sciences. The enantiomers of organic compounds can exhibit very different pharmacological activities, although they have the same chemical structure [1]. The pharmacodynamic and pharmacokinetic differences between enantiomers can result in stereoselective toxicity. For example, L-dopa is very effective in treating Parkinson’s disease, while D-dopa can cause serious side effects. Furthermore, for thalidomide, the *R*-(+)-enantiomer is an effective sedative, while the *S*-(−)-enantiomer can cause fetal abnormalities. Therefore, chiral analysis of individual enantiomers is essential for both fundamental and applied science. It is very important to promote the chiral analysis and separation of racemic drugs to eliminate the unwanted isomer and to find an optimal treatment in clinic. Various techniques have been developed and applied for this goal, including chromatography, circular dichrosim, nuclear magnetic resonance and mass spectrometry (MS) [2,3,4]. For example, chiral chromatography including high performance liquid chromatography (HPLC) and gas chromatography (GC) has been widely used in this field [1,2,3,4]. Despite this, the development of more efficient and rapid techniques for chiral analysis and separation is still greatly desired.

Since two enantiomers have same mass and very similar properties, MS was thought to be chiral-blind in the early years. It was realized later that if a chiral environment could be introduced to enantiomers in the gas phase, chiral differentiation could be fulfilled by MS-relative methods. Since the electrospray ionization (ESI) method can be used to generate unstable noncovalent complexes readily, it has become the best partner of the MS-based chiral analysis method and greatly promoted the development of the research field since its invention [5,6,7,8,9,10]. In addition, it makes the MS-based chiral analysis a very attractive method compared with others due to its excellent performance in terms of rapidity, sensitivity, and specificity [9,10].

Due to the poor reproducibility of single-stage MS, nowadays the MS-based chiral analysis mostly depends on multiple-stage MS methods, including tandem MS, ion mobility MS, and photodissociation of selected ions. Among them, the method of infrared multiple photon dissociation (IRMPD, or infrared photodissociation, IRPD) spectroscopy has shown its unique advantages in the issue [11,12,13,14,15,16,17,18,19]. In addition to realizing chiral discrimination, infrared photodissociation IRMPD spectroscopy can further provide spectral and structural information of chiral complexes, such as charge position, hydrogen bond interaction, and kinetic information. The IRMPD method has been applied in many different molecular systems, including amino acids, peptides, DNA bases, carbohydrate, metabolites, and small molecules [13,14,15,16,17,18,19]. In this review, we will only focus on its application in chiral analysis. The application examples are summarized and its prospects are also discussed.

## 2. IRMPD Spectroscopy: A Brief Description of the Technique

The premise of using IRMPD spectroscopy for chiral analysis in the gas phase is to provide an effective chiral environment for the chiral molecules to be analyzed, which usually requires forming diastereomeric complexes, that is, forming diastereomeric complexes first. Although the neutral molecule beam method is also applied in chiral analysis [20,21,22,23,24], the more widely applied method is the one to obtain the IRMPD spectrum by recording the fragment ions induced by the irradiation of tunable IR laser on the mass-selected complex ions in an ion trap. In this review, we will only focus on the latter for the sake of clarity.

Figure 1 shows a typical schematic diagram of this method. After selecting suitable chiral reference molecules and metal ions (if necessary) for the chiral analyte, the diastereomeric complex ions can be formed by ESI for the enantiomers, respectively. The target complex ions are then mass-selected and trapped in an ion trap. After that, a tunable IR laser is introduced into the trap to irradiate the target ions for a given time. If the diastereomeric ions have different IR absorption properties or different dissociation kinetics, their product mass spectra after the IR irradiation can be differentiated by their fragmentation patterns or precursor ion intensities. The spectral intensity at each wavenumber can be calculated as *I_in_* = −*ln* (I_p_/(ΣI_f_ + I_p_)), where the intensities of parent ions and fragment ions are identified by I_p_ and I_f_, respectively. In cases without observed fragment ions, the spectral intensity can be calculated as *I_in_* = −*ln* (I’/I_0_), where the intensities of precursor ions before and after IR irradiation are identified by I_0_ and I’, respectively. Due to the different structures of the diastereomeric complex ions, their IRMPD spectra can be differentiated by the positions, shapes, and relative intensities of some characteristic spectroscopic peaks. Therefore, by analyzing the IRMPD spectra of the diastereomeric complex ions, the chiral distinction can be achieved. Further combined with theoretical calculations, their difference in structure, interaction, and dynamic kinetics can be further discovered.

Experimentally, the IRMPD spectroscopy is usually implemented on Fourier transform ion cyclotron resonance (FT ICR) or ion trap mass spectrometers equipped with ESI sources, combined with different types of tunable IR laser. The output wavenumbers can cover the range of 500–2000 cm^−1^ or 2600–4000 cm^−1^ (or parts of them), depending on the laser applied in the experiments. For example, the powerful free electron laser for infrared experiments (FELIX) in the Netherlands and Centre Infrarouge Laser Orsay (CLIO) in France have been applied in the relative studies and have provided lots of valuable experimental results [15,16]. On the other hand, the bench top optical parametric oscillator/amplifier (OPO/A) laser is also a good choice for most laboratories [13,14,15,16,17,18]. The irradiation time can be controlled by a mechanical shutter in the light way or the designed trigger for the laser. Usually the experiments are performed at room temperature (~293 K). However, the experiments can also be performed under a temperature below 10 K with a special cooling design on the ion trap. A typical experimental setup combing a 7T FT ICR mass spectrometer and a tunable OPO laser in the region of 2700–4000 cm^−1^ is shown in Figure 2.

## 3. Application of IRMPD Mass Spectrometry and Spectroscopy in Chiral Analysis

### 3.1. Calixarene Complexes

As the first example of IRMPD spectra of diastereomeric complex ions, Filippi et al. explored the effect of chirality on the spectra of diaminoresorcinol-[4]calixarene complexes [25,26]. They chose the *R*, *R*, *R*, *R*-(R) and *S*, *S*, *S*, *S*-(S) enantiomers of diaminoresorcin-[4]calixarene as the chiral acceptors for the chiral guests of cytidine and its epimer cytarabine. The protonated ions were generated by ESI, trapped in an ion trap and irradiated by a focused IR laser with a tunable range of 2800–3600 cm^−1^ through a conical hole in the ring electrode. As shown in Figure 3, both complex ions of [R + cytidine + H]^+^ and [S + cytidine + H]^+^ show sharp peaks at 3425 cm^−1^, broad peaks in the range of 3100–3300 cm^−1^, and some peaks in the range of 2960–3100 cm^−1^. However, some differences in the signal intensity and peak shape can be observed in their spectra. In contrast, the spectra of cytarabine are much different. For both ions of [R + cytarabine + H]^+^ and [S + cytarabine + H]^+^, the peaks in the 2960–3100 cm^−1^ have quite different relative intensity distributions and shapes, although they still have the same positions. The sharp peak at 3425 cm^−1^ has slightly blue-shifted to 3420 cm^−1^. For [R + cytarabine + H]^+^, the broad peak has red-shifted to the region of 3200–3300 cm^−1^; while for [S + cytarabine + H]^+^, its intensity has been decreased manifestly. Remarkably, a pronounced signal at 3354 cm^−1^ was only observed for the latter. Further calculations show that their IRMPD spectra agree with the coexistence of several isomeric structures, characterized by the protonated guest accommodated inside or outside the host cavity. Furthermore, the differences in their spectroscopic patterns reflect the effect of intramolecular hydrogen bonding on the structures of these diastereomeric complexes.

### 3.2. Serine Octamer Clusters

This protonated serine octamer was first observed as a magic cluster ion in a mass spectrometer by Cooks et al. in 2001 [27]. Very significantly, the magic cluster ion also shows great homochiral advantage. Further experiments show that the homochiral advantage of serine octamer can be transferred to other amino acids or carbohydrates through enantioselective substitution reactions [28,29,30,31,32]. Considering that the phenomenon of self-disproportionation of enantiomers (SDE) has a significant effect on homo- vs. heterochiral associations, the serine octamer may play an important role in chiral transmission and chiral enrichment [31,32,33,34]. The method of IRMPD spectroscopy has been applied to study the structure of the magic clusters by several groups and has provided valuable information for a better understanding of its structures and the significant homochiral advantage [13,33,34,35,36,37,38,39].

Serine octamer ions have been also applied for chiral analysis for of its component units. Xu and his colleagues applied IRMPD spectroscopy to study the chiral discrimination of protonated serine octamers in the wavenumber range of 3200–3800 cm^−1^ [39]. As shown in Figure 4, the IRMPD spectra of homochiral (8*S*) and heterochiral (7*R* + 1*S*) serine octamers show differences in the region of 3250–3500 cm^−1^. The homochiral cluster has a broad peak around 3410 cm^−1^ and a relatively narrow peak at 3330 cm^−1^, which are not reflected in the heterochiral one. Furthermore, the two peaks observed at 3556 and 3674 cm^−1^ for the homochiral cluster are also weakened for the heterochiral one.

Since one or two units in serine octamers can be easily substituted by other amino acids with the ESI source, Kong’s group investigated the substituted serine octamer systems based on IRPD mass spectrometry and spectroscopy [40,41,42]. The IR dissociation mass spectra of the proline-substituted serine octamer is shown in Figure 5. For the mono-substituted product, the dissociation pathways and products of the homochiral and heterochiral clusters are quite similar. However, for the di-substituted ions, their dissociation paths are significantly different. According to their difference in the mass spectra, it is easy to realize the chiral distinction between homochiral and heterochiral di-substituted octamer clusters. For the threonine-substituted serine octamer, similar results were observed. On the other hand, chiral discrimination of these substituted serine octamer clusters of [l-Ser_7_ + l/d-Pro_1_]H^+^, [l-Ser_6_ + l/d-Pro_2_]H^+^, [l-Ser_7_ + l/d-Thr_1_]H^+^ and [l-Ser_6_ + l/d-Thr_2_]H^+^ can also be achieved by comparing their IRPD spectra. Figure 6 shows some of the results. For a clearer comparison, the spectrum of [l-Ser_8_]H^+^ obtained on the same experimental instrument is also shown in the figure. All the clusters have broad absorptions in the region of 2800–3250 cm^−1^, but the spectra can be differentiated in the region of 3300–3550 cm^−1^. Chiral differentiation of substituted serine octamers can be achieved by comparing their IRMPD spectra in this region. Taking threonine-substituted serine octamers as the examples, the peak at 3425 cm^−1^ corresponding to the H-bonded aliphatic O–H, does not change much for [l-Ser_7_ + l-Thr_1_]H^+^ compared to [l-Ser_8_]H^+^, but broadens for [l-Ser_7_ + d-Thr_1_]H^+^. This trend becomes clearer when two units of L-Ser were replaced. For [l-Ser_6_ + d-Thr_2_]H^+^, the peak shifts to 3465 cm^−1^ clearly, very different from that of the [l-Ser_6_ + l-Thr_2_]H^+^. Interestingly, similar results have also been observed for the proline and phenylalanine substituted serine octamers [40,41,42].

### 3.3. β-Cyclodextrin Complexes

Cyclodextrins (CDs) are closed cyclic oligosaccharides, which are wide at the top and narrow at the bottom in their structures [43,44]. According to the number of oligosaccharides in the molecule, cyclodextrins can be divided into three types: α (6), β (7), and γ (8). The inner cavity of cyclodextrin is hydrophobic due to the exposed CH groups, while the outer surface is hydrophilic due to the OH groups protruding from the edge. Its special structure allows it to form host and guest complexes with a variety of small molecules and has been widely applied in the field of chiral analysis and separation [45,46,47,48,49].

Based on IRMPD spectroscopy and density functional theory (DFT) calculations, Oh et al. studied the chiral discrimination of d/l-alanine and d/l-isoleucine by permethylated β-CD [50,51]. With a FT ICR mass spectrometer combined with OPO lasers, the experimental IRMPD spectra of the complexes formed by permethylated β-CD with d/l-alanine were obtained in the range of 2600–3800 cm^−1^. Remarkably, the two spectra show great difference. As shown in Figure 7, except for the peak at 3008 cm^−1^, the two spectra have no common band in the recorded spectral region. Further DFT calculations show that the interaction between these two molecules of permethylated β-CD and alanine makes significant structural differences in the solvent-free gas phase environment. Similarly, chiral differentiation is fulfilled in the systems of permethylated β-CD and d/l-isoleucine [51]. The spectrum of complex ions with d-IIe in the low wavenumber region shows two weak peaks at 2750 and 3125 cm^−1^, while that for l-IIe shows two different peaks at 2869 and 3003 cm^−1^. In the high spectral region of 3370–3565 cm^−1^, both samples have three adjacent peaks, but the peak positions are still quite different. Their detailed calculations show that the difference is mainly due to the local interaction between the protonated amino acids and permethylated β-CD. In addition to exploring the chiral identification with permethylated β-CD, β-CD is also applied in such kinds of research. Sun et al. used the method to study the chiral differentiation of penicillamine (Peni) [52]. The chirality of l/d-Peni also leads to a significant difference in the region of 2900–3400 cm^−1^ in their IRMPD spectra.

Very recently, Hirata et al. studied the complexes of permethylated β-CD with the two enantiomers of protonated tyrosine in a different way [53]. The target host–guest complexes were generated, mass-selected, and introduced to a cryogenic quadrupole ion trap at 4 K. The H_2_-tagged complex ions were then formed by the introducing of the buffer gas including H_2_. Both their IR spectra in the regions of 1100–1900 cm^−1^ and 2700–3700 cm^−1^ are recorded, and results show that their spectra are quite different in the regions of 1100–1300 and 3000–3400 cm^−1^. It is concluded that the specific interactions are responsible for the recognition process.

### 3.4. Compexes with Axially Chiral Ligand

After realizing the chiral distinction of diaminoresorcinol-[4] calixarene enantiomers R and S, Filippi et al. later carried out another interesting work—IRMPD spectral identification of diastereomeric complexes using an axially chiral ligand [54]. They used the axially chiral multifunctional macrocycle molecule of M^aR^ to form protonated complexes with l-/d-dihydroxyphenylalanine (D^L^/D^D^) through the ESI method. By recording the intensity of ionic fragment of protonated M^aR^ upon tuning the IR laser, the IRMPD spectra were obtained in the range of 2800–3700 cm^−1^. From the results (Figure 8), it is found that the spectrum of [M^aR^·H·D^D^]^+^ is quite different from that of its [M^aR^·H·D^L^]^+^ diastereomer. The different spectral signatures are mainly from NH and OH stretch regions. The comparison of the experimental spectra with the calculated ones indicates that these characters can be attributed to the coexistence of several stable rotamers generated in the ESI source.

### 3.5. Dimers of the Chiral Molecules

In addition to using special chiral molecules as chiral references, the chiral analyte (and its enantiomers or its close derivatives) can also be applied to form the diastereomeric complexes. The protonated or metal cationized homochiral or heterochiral dimer can be generated in the ESI and studied by IRMPD spectroscopy for the goal of chiral analysis.

For example, Klyne et al. used gas phase vibrational spectroscopy to study the chiral recognition of protonated glutamate dimers [55]. In their experiments, the ESI-generated protonated homo- and hetero- dimers of glutamate was mass-selected and trapped in a quadrupole ion trap held at 10 K before the form of cold clusters of ll-/ld-Glu_2_H^+^-H_2_ through the collision with the He/H_2_ buffer gas. Then, the IRPD spectra of H_2_-tagged ll-/ld-Glu_2_H^+^-H_2_ dimers were recorded in both ranges of 1100–1900 cm^−1^ and 2600–3600 cm^−1^. As shown in Figure 9, the results show clear spectroscopic signatures with different peak positions and intensities. The NH bending modes (bands N, M and L) and CO stretches (bands K1–K4) displayed in the fingerprint range are found to be sensitive to the chirality. For their IR characteristics in the NH and OH stretch ranges, significant difference between them can be identified too. It is believed that these discrepancies result from their conformational landscapes, including the most stable structures, conformer populations and interaction types and strengths. Thus, the author further studied the stereochemistry-induced effects with the aid of other experimental methods and quantum chemistry. Some very interesting results were revealed, such as the coexistence of the multiple conformers and the fact of the incomplete thermalization in the cold trap.

The IRMPD spectra of the protonated homochiral serine dimer and heterochiral serine dimer were studied by Sunahori et al. [39]. It was found that the spectrum of the heterochiral cluster is identical to the one of homochiral cluster in the 3400–3800 cm^−1^ region, but is significantly lower in intensity in the 3200–3400 cm^−1^ region. On the other hand, a theoretical study was carried out for such protonated dimers for predicting the performability of IRMPD spectroscopy on the chiral differentiation for amino acids. Using density functional theory (DFT) calculations and molecular dynamics simulations, Poline et al. studied the infrared characteristics of homochiral and heterochiral protonated amino acid dimers (Ala_2_H^+^, Ser_2_H^+^, Thr_2_H^+^, Phe_2_H^+^, Arg_2_H^+^) [56]. The results predict that these dimers, except Ala_2_H^+^, have distinct mid-IR absorption bands in homochiral and heterochiral configurations, making them appropriate to be studied with mid-IR photon dissociation spectroscopy.

Barbu–Debus et al. studied the homochiral and heterochiral sodium core dimers of tartaric acid esters [57]. The dimers of dimethyl-l-tartrate (Lmt) and the two enantiomers of diisopropyl tartrate (Lipt and Dipt) are generated by ESI and studied by IRMPD method in both fingerprint and 3 µm regions. Results show that no chiral effect observed in the fingerprint area for [Lmt + Lipt + Na^+^] and [Lmt + Dipt + Na^+^]. However, a slight difference exists in the OH stretch area, in which a shoulder peak only appears in the broad absorption peak of the homochiral dimer. The calculation result indicates that the formed gas phase complexes probably reflect the pre-existing solution structures.

### 3.6. Molecules of Diastereomers

For the differentiation of enantiomers that mirror each other by IRMPD spectroscopy, the creation of a chiral microenvironment by introducing a chiral reference is necessary. However, for the diastereomers with two or more chiral centers, the chiral reference is not necessary since their structural difference might be reflected by IRMPD directly. To form stable ions of the diastereomers, protonation or metal cationization is usually needed. In many cases, the metal cationization can enlarge the chiral difference and make the formed cationized stereoisomers differentiated more easily.

Dunbar et al. studied the IR ion spectroscopy of M^+^-PhePhe complexes (M = Li, Na, H) for both LL and DL stereoisomers of the dipeptide in 2010 [58]. The ESI-generated complex ions were isolated in a FT ICR cell and irradiated with the tunable output of the FELIX IR laser. As shown in Figure 10, the protonated ions show no obvious spectral difference, while the alkali metal cationized ones show significant differences in the carbonyl stretching region for LL and LD forms. Further analysis indicates that the observed chiral discrimination originates from the intramolecular interaction remote from the metal-binding site, instead of the site itself.

Crestoni et al. studied the RRMPD spectra of protonated diastereomers (2*S*,4*R*)-4- hydroxyproline (HypH^+^) and (2*S*,4*S*)-4-hydroxyproline (hypH^+^) in both regions of 950−1950 cm^−1^ and 3200−3700 cm^−1^, using the CLIO FEL and their tabletop IR laser coupled with ion trap mass spectrometer [59]. Results show remarkable differences in the fingerprint region, including the carbonyl stretching modes existing at 1750 and 1770 cm^−1^ for *S*,*R* and *S*,*S* diastereomers, respectively. In addition, the NH_2_ wagging mode at 1333 cm^−1^ is a distinct mark of the *S*,*S* isomer. In the NH/OH stretching region, the two species share similar absorptions, accompanied with the subtle difference in the weak bands at 3320 and 3336 cm^−1^ for HypH^+^ and hypH^+^, respectively. Further calculations show that each protonated epimer comprises at least three conformers, which are stabilized by intramolecular hydrogen bonds.

Alata et al. studied the structure of the protonated dipeptide of cyclodiphenylalanine with the methods of IRMPD spectroscopy and quantum chemical calculations [60]. Experimental IRMPD spectra of protonated species of cyclo LPhe-DPhe (c-LDH+) and cyclo LPhe−LPhe (cLLH+) were studied by a 7 T FT ICR hybrid mass spectrometer coupled to the CLIO FEL or a tabletop OPO laser. In the fingerprint region, the main differences between the two diastereomers is the blue shift of several bands of c-LLH^+^ relative to those of c-LDH^+^. In the ν(OH) stretch region, some differences at 3592 cm^−1^ can be identified.

Lepere et al. studied the structures of protonated and sodiated polyphenylalanine (dipeptide and tetrapeptide) with IRMPD spectroscopy in the 900–2000 cm^−1^ region combined with other methods and calculations [61]. For the protonated LL and LD dipeptides, the only difference is that the LL spectrum shows a higher intensity than that of LD at 1530 cm^−1^. For the sodiated ions, the main difference is that the LLNa^+^ spectrum shows a shoulder peak at 1780 cm^−1^, which is not present in that of LDNa^+^. Further studies have shown that the LLNa^+^ has two kinds of isomers with different spectral characteristics, and LDNa^+^ only exists in one kind of isomer. Furthermore, the competition between Na^+^···O and Na^+^···π interactions strongly depends on chirality. For protonated tetrapeptides, Figure 11 shows the experimental spectra of protonated LLLL and LDLD. Compared with LLLLH^+^, the NH bending region and CO stretching region of LLDDH^+^ have fewer structural features in the NH bend and CO stretch regions. In addition, LDLDH^+^ shows the presence of a band at 1241 cm^−1^ and the absence of a band at 1141 cm^−1^ relative to LLLLH^+^. These spectral features are explained by their different secondary NH^+^···π interactions. However, for the sodiated ions, their spectra differ only in the relative intensity of the 1680 cm^−1^ peak, indicating that the spectral characteristics are not strongly affected by the residue chirality in the sodiated tetrapeptides.

## 4. Discussion

The method of IRMPD spectroscopy performs chiral differentiation by the analysis of the IRMPD spectra of the formed diastereomeric ions. It is based on tandem mass spectrometry, and keeps its advantage of sensitivity and specificity. The unique advantage of the method is that it provides valuable structural information of chiral complexes. When combined with high-level calculations, the chiral differentiation can be understood on the fundamental level and provides new hints for the design of more effective chiral references for analysis or ligands for separation. The spectroscopic method is sensitive to structures, thus the structurally different diastereomeric complex ions can be differentiated effectively by their IRMPD spectra even when they have very close binding energies. In most cases, the structural differences for the diastereomeric ions are induced by intermolecular (or intramolecular) interactions, such as hydrogen bonding, M^+^···π and π···π interactions [61,62].

However, the method also brings some disadvantages in chiral analysis. For one thing, it is not so easy to obtain the infrared spectra, and higher requirements are put forward for the experimental instrument. Secondly, the co-existing isomers make spectra analysis difficult. Experiments and theoretical calculations have revealed that even at a temperature of (or below) 10 K, the coexistence of multiple isomers for protonated amino acid clusters and other noncovalent complex ions is still inevitable [55,63]. This fact makes a single experimental spectrum to be a sum of multiple spectra from different isomers, complicating the problem and making the quantitative chiral analysis based on the method difficult to be achieved. Considering the ratios of the coexisting multiple isomers of the noncovalent complexes generated in the ESI source might vary according to experimental conditions, such as solvent, temperature and voltage, the application of the IRMPD method in the field should be considered and designed very carefully [64,65].

Thus, it can be suggested that some more rigid and stable diastereomeric ions generated for both enantiomers might be a good choice for IRMPD experiments. It is well known that in MS/MS- based approaches, chiral recognition can be determined by comparing the intensity ratios of the product ions or the product and precursor ions after the process of collision-induced dissociation (CID) on the complex ions [66,67,68,69,70,71,72]. Typically, the applied diastereomeric complex ions have a form of [(M^II^) (A)(Ref)_2_−H]^+^, which are formed by chiral analytes (A), chiral reference compounds (Ref) and metal ions (M) [66]. Because of the coordination of metals, these complex ions can be readily generated by ESI source and are quite stable in the gas phase. We are thinking whether these complex ions are good choices for IRMPD-based chiral analysis, the ions of [(Cu^II^) (d/lTyr)(l-Pro)_2_−H]^+^ is selected here as an example, in which the l-Pro is the reference compound, and the analytes are l- and d- Tyr. The experiments are performed in the range of 2700–3750 cm^−1^. However, both spectra show strong absorption at the same positions although their relative intensities at certain wavenumbers are some different (data not shown here). The results can be rationalized by the fact the strong coordination greatly breaks most of the hydron bonds and thus means the spectral bands are very similar to each other and can be hardly differentiated according to their position. However, the results also encourage us to think of the problem in a different way: if multiple hydrogen bonds can form in the outer layer of the coordination center, the structure difference might be reflected by their IR spectra governed by their different hydrogen bonds. Thus, a different complex ion of [(Cu^II^) (Tyr)(l-Pro)_3_−H]^+^, which is also stable and has high abundance in the ESI mass spectra, is selected and applied to the IRMPD spectroscopy study. As shown in Figure 12, the idea is confirmed by their experimental IRMPD spectra. The results can help us to design IRMPD experiments for the goal of chiral analysis in a more reasonable way, by choosing the most appropriate chiral reference (and metal ions, if necessary) to get the balance between stabilizing the diastereomeric ions in energy (to reduce the possibilities of the existed isomers) and differentiating them in structures to highlight their spectral differences.

## 5. Conclusions and Future Directions

This article reviews the chiral recognition research of IRMPD spectroscopy applied to different molecular systems, and shows that IRMPD spectroscopy is an effective chiral analysis tool. The method is complementary to other relative methods by providing the valuable IR spectra of diastereomeric ions that can be readily generated by ESI source. The experimental spectra can be compared with high-level calculations to reveal the chiral difference at the molecular level. However, the IRMPD spectroscopy technique still has some limitations. The extensive coexistence of multiple isomers of the complex ions makes the interpretation and quantitative comparison of spectra difficult. Thus, the design of the chiral reference is of great importance to develop its practical value in chiral analysis. On the other hand, the types of fragment ions provided by IRMPD are same as those observed in CID experiments. Some recent research results show that UV photodissociation (UVPD) can generate fragment ions with very high diversity in species and can be applied in the chiral analysis effectively [73,74,75,76,77]. Thus, it is more promising to explore how IRMPD can be further combined with other techniques in such research. At the same time, more effective high-level calculations for structure searching is also very helpful to better understand the molecular basis of chiral recognition.

## Figures and Tables

**Figure 1 molecules-25-05152-f001:**
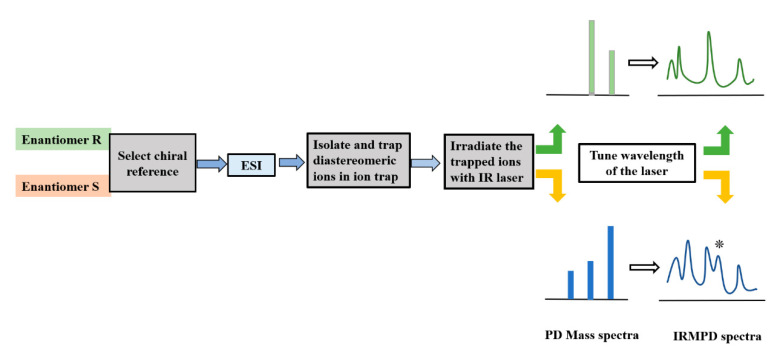
A typical schematic diagram to show the application of infrared multiple photon dissociation (IRMPD) spectroscopy in chiral analysis.

**Figure 2 molecules-25-05152-f002:**
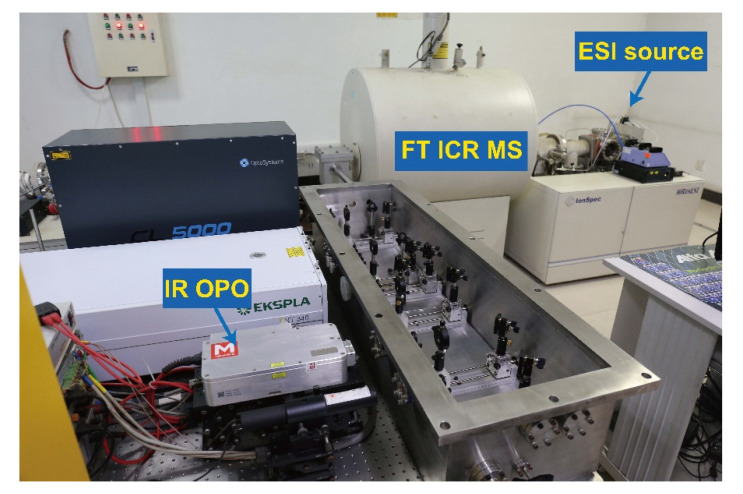
The experimental setup for IRMPD experiments in Nankai University, based on a 7 T FT ICR mass spectrometer (Ionspec) and a tunable optical parametric oscillator (OPO) laser (Msquares).

**Figure 3 molecules-25-05152-f003:**
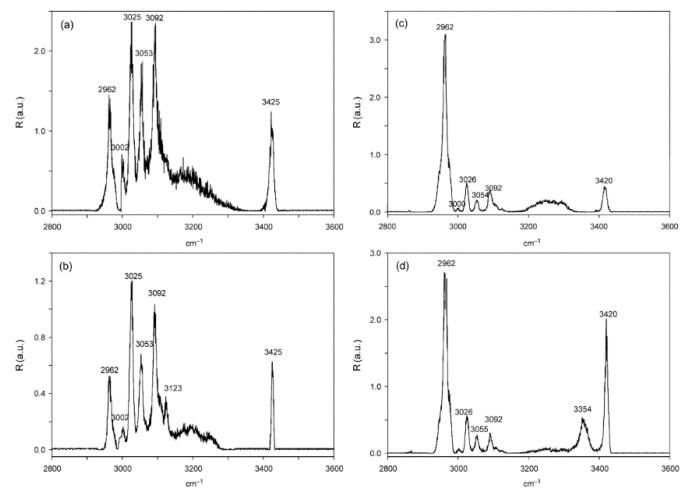
IRMPD spectra of the ESI-formed diastereomeric complexes: (**a**) [R + cytidine + H]^+^; (**b**) [S + cytidine + H]^+^; (**c**) [R + cytarabine + H]^+^ and (**d**) [S + cytarabine + H]^+^ [25]. Reprinted with permission from Wiley.

**Figure 4 molecules-25-05152-f004:**
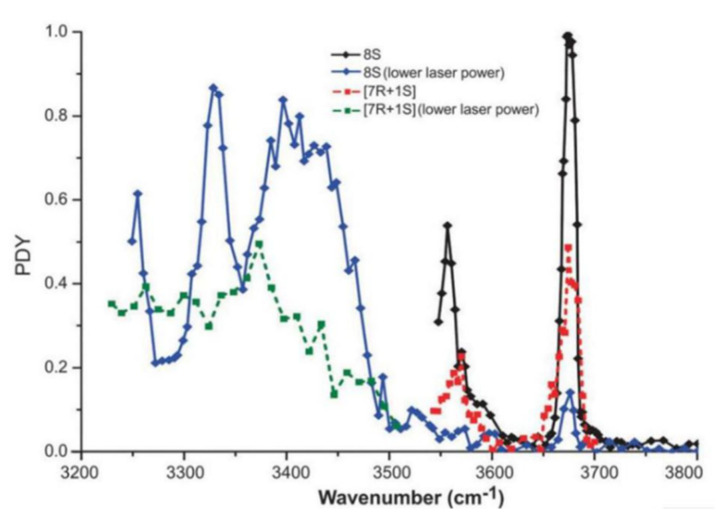
IRMPD spectra of the homochiral protonated serine octamer [8S] (black and blue, solid) and the heterochiral protonated serine octamer [7*R* + 1*S*] (red and green, dotted). The black [8*S*] and red [7*R* + 1*S*] traces were measured with a higher laser power compared to the blue and green traces [39]. Reprinted with permission from RSC.

**Figure 5 molecules-25-05152-f005:**
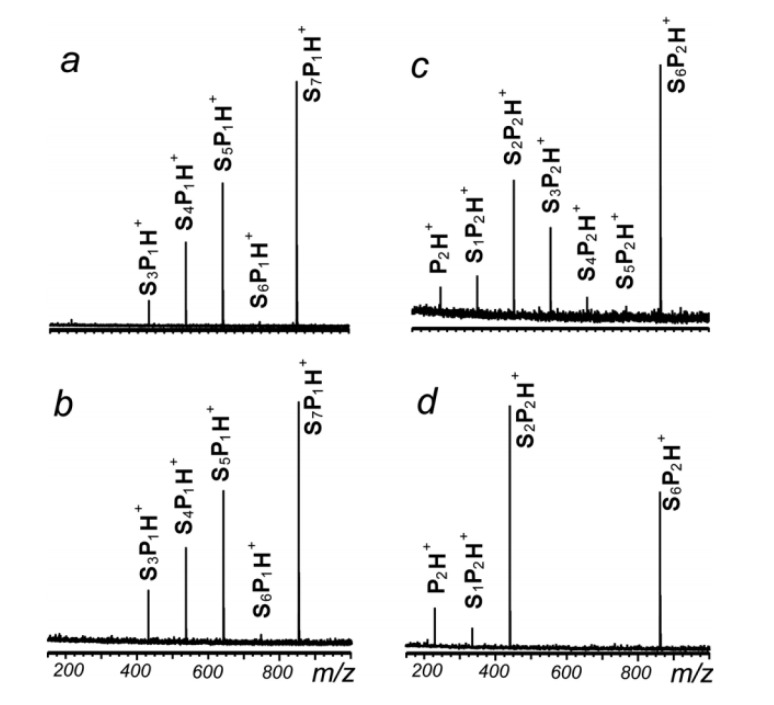
IRMPD mass spectra of (**a**) [l-Ser7 + l-Pro1]H^+^; (**b**) [l-Ser7 + d-Pro1]H^+^; (**c**) [l-Ser6 + l-Pro2]H^+^ and (**d**) [l-Ser7 + d-Pro2]H^+^ [40]. Reprinted with permission from RSC.

**Figure 6 molecules-25-05152-f006:**
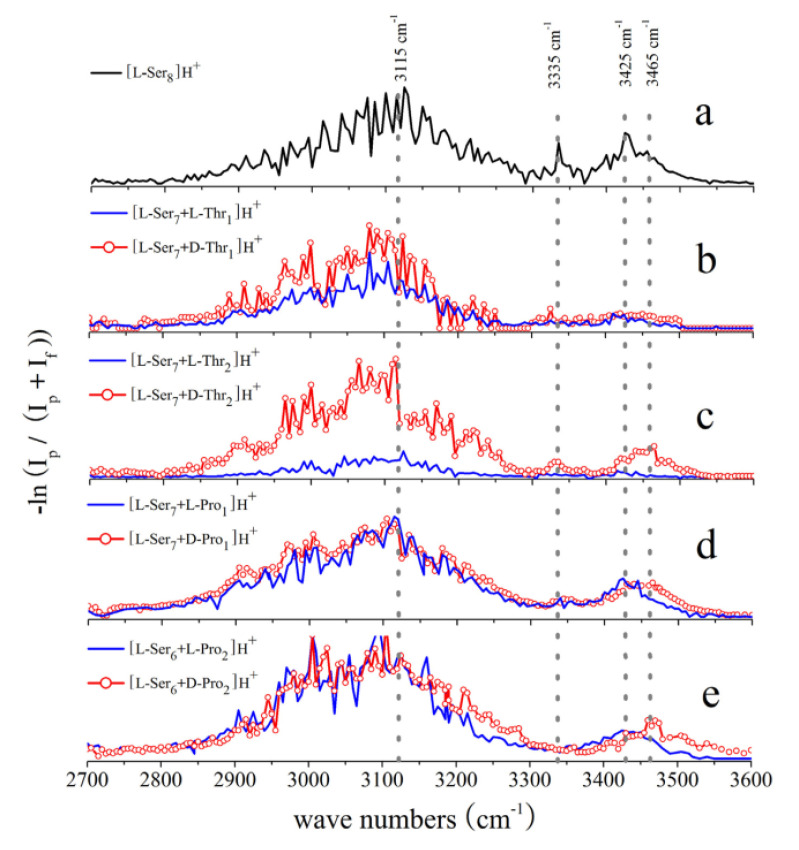
IRMPD spectra of (**a**) [l-Ser_8_]H^+^; (**b**) [l-Ser_7_ + d/l-Thr_1_]H^+^; (**c**) [l-Ser_6_ + d/l-Thr_2_]H^+^; (**d**) [l-Ser_7_ + d/l-Pro_1_]H^+^, and (**e**) [l-Ser_6_ + d/l-Pro_2_]H^+^ [41]. Reprinted with permission from Elsevier.

**Figure 7 molecules-25-05152-f007:**
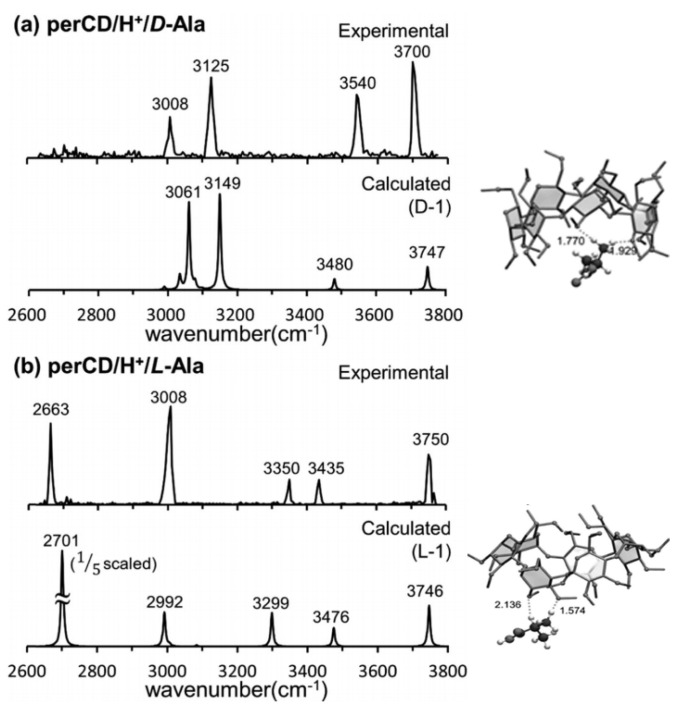
Experimental IRMPD and calculated absorption spectra of the protonated complex ions of permethylated CD (perCD) and (**a**) d-Ala and (**b**) l-Ala [50]. The structures of calculated isomers of D-1 and L-1 with the chiral reference of perCD and enantiomers of d/l-Ala are shown in the right, respectively. Reprinted with permission from RSC.

**Figure 8 molecules-25-05152-f008:**
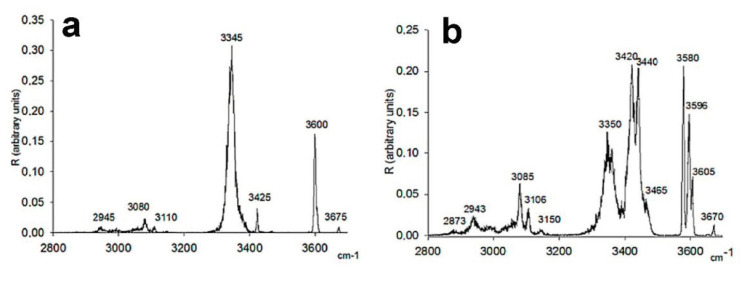
IRMPD spectra of (**a**) [M^aR^·H·D^L^]^+^ and (**b**) [M^aR^·H·D^D^]^+^ [54]. Reprinted with permission from Wiley.

**Figure 9 molecules-25-05152-f009:**
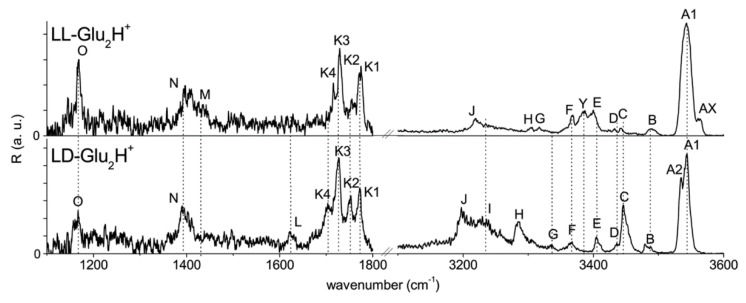
Overview IRPD spectra of diastereospecific ll- (**top**) and ld-Glu_2_H^+^-H_2_ (**bottom**) recorded in the XH stretch (X = O, N; 3100–3600 cm^−1^) [55]. Reprinted with permission from RSC.

**Figure 10 molecules-25-05152-f010:**
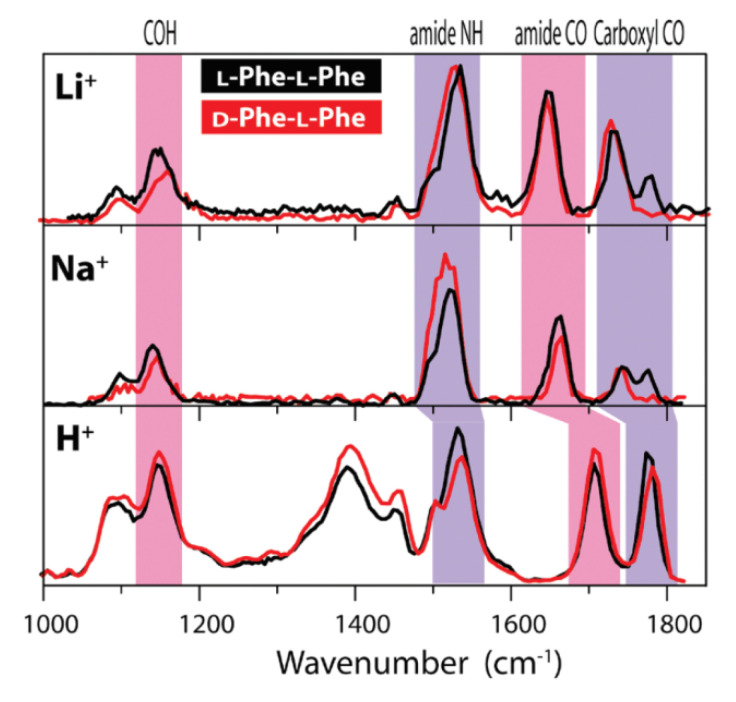
Experimental IRMPD spectra of (**top**, **middle**) alkali metal ion complexes with the LL and DL stereoisomers of the dipeptide PhePhe and (**bottom**) the protonated LL and DL dipeptides. The splitting of the carboxyl CO stretching peak into two components in the LL complexes (black traces), which is evident for Na^+^ and Li^+^, reflects the nearly equal stabilities of two spectrally distinct conformers. For the DL dipeptide (red traces), only one of the conformeric motifs is formed with Na^+^ and Li^+^, as indicated by the complete absence of one of the carboxyl CO stretching bands [58]. Reprinted with permission from ACS.

**Figure 11 molecules-25-05152-f011:**
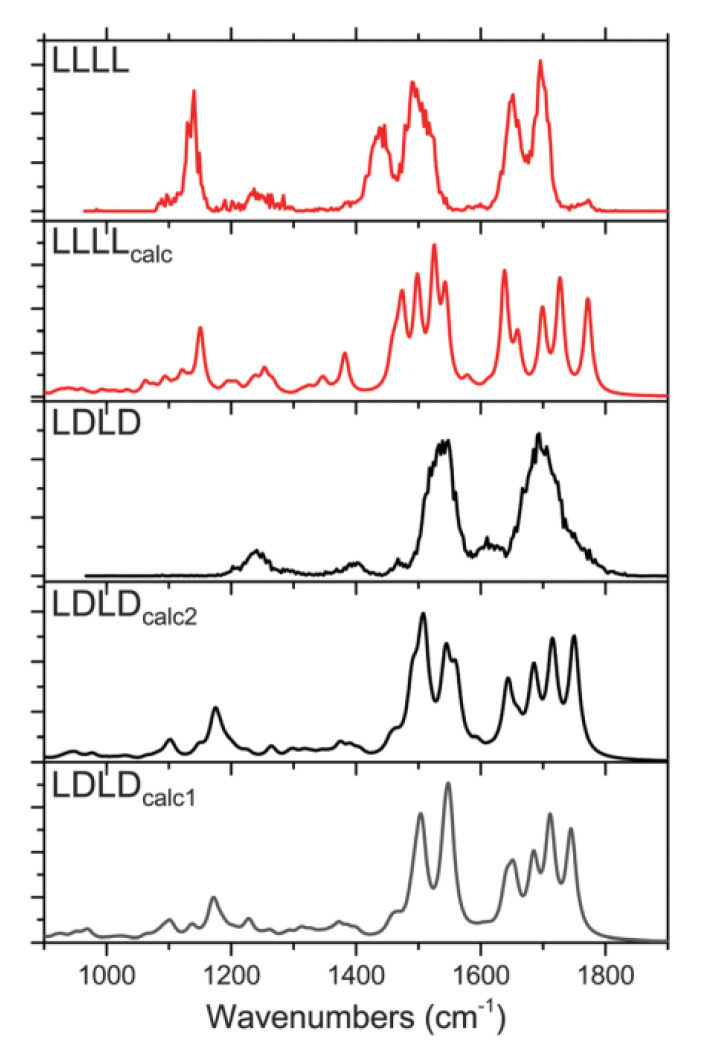
Experimental spectra of protonated LLLL and LDLD together with some simulated spectra (LLLL_calc_, LDLD_calc1_ and LDLD_calc2_) [61]. Reprinted with permission from RSC.

**Figure 12 molecules-25-05152-f012:**
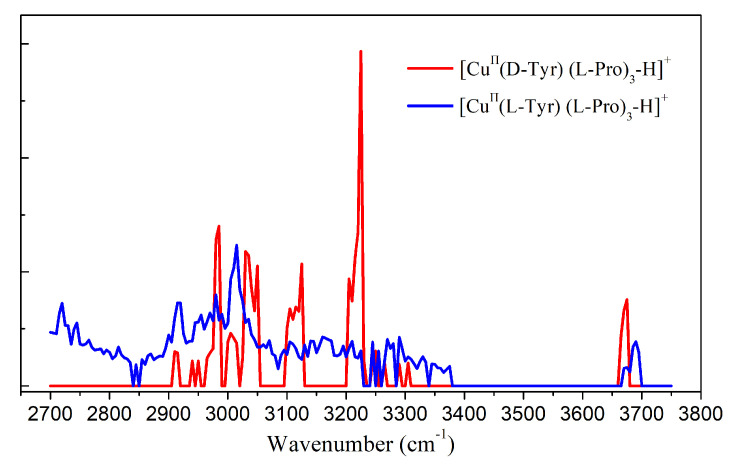
Experimental IRMPD spectra of [(Cu^II^) (d/l-Tyr)(l-Pro)_3_−H]^+^.

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
