# Peer review of "Application of Infrared Multiple Photon Dissociation (IRMPD) Spectroscopy in Chiral Analysis"

_molecules, 2020, doi:10.3390/molecules25215152_

Round 1
Reviewer 1 Report
Infrared multiple photon dissociation (IRMPD) spectroscopy is a very an attractive in the field of chiral analysis. Several chiral molecules, including amino
acids and peptides, have been studied with the method combined with theoretical calculations. The method of IRMPD spectroscopy performs chiral differentiation by the analysis of the IRMPD spectra of the formed diastereomeric ions. The review presents experimental results and discuss limitation and prospects of the method.
In the Introduction section a more comprehensive presentation of chiraility and its implication in therapy is needed. Also a short presentation of the most common chiral separation methods would be useful.
The language of the article can be improved, there are several grammar and syntax errors in the manuscript.
The advantages and disadvantages of the method should be briefly presented in the Discussion section.
Is the technique used in combinations with other methods for chiral analysis ?
On which mechanism is chiral recognition based ?
Reviewer 2 Report
The review is good and summarizes the new and exciting field of research, yet of limited general interest. Well-written and presented; The authors should include work by Professor M. Suhm, in particular: "Chirality-dependent sublimation of a-(trifluoromethyl)-lactic acid: Relative vapor pressures of racemic, eutectic, and enantiomerically pure forms, and vibrational spectroscopy of isolated (S,S) and (S,R) dimers", J. Fluor. Chem. 2010, 131, 495-504. Also, the phenomenon of Self-Disproportionation of Enantiomers should be discussed in general, as it has significant effect on homo- vs. heterochiral associations, in particular in the case of serine octamer clusters.
Reviewer 3 Report
The review article of Kong and co-workers describes recent applications of Infrared Multiple Photon Dissociation (IRMPD) spectroscopy in chiral analysis. The manuscript is well and clearly written and in my opinion understandable for researchers not familiar with this technique (methodology). The work helps in understanding of application of IRMPD spectroscopy in chiral analysis. I have only few minor remarks which may help authors improve the impact of their manuscript:
1) the recent review P. Maitre, et al. Applications of Infrared Multiple Photon Dissociation (IRMPD) to the Detection of Posttranslational Modifications, Chem. Rev. 2020, 120, 3261−3295 should be inserted in the reference list;
2) Keyword: “Chiral” should be changed to “Chirality”;
3) the title of paragraph 2 is not really strong: I suggest to change “Experimental methods” to “IRMPD spectroscopy: a brief description of the technique”;
4) Paragraph 3: for clarity molecular structures of enantiomers and/or chiral references should be inserted close to the reported IRMPD spectra;
5) in my opinion paragraph 4 should be included in the final paragraph “Conclusion and future directions”
Author Response
Please see the attachment。

Round 2
Reviewer 1 Report
The authors complied with the suggested modifications.
Chapter 4 should be remained from "Some Discussion" to Discussion
Examples of chiral pharmaceuticals and the implication of stereochemistry in therapy can be introduced in the Introduction section
Author Response
The authors thank the comments and suggestions made by the reviewer very much. According to these comments and suggestions, we made corresponding corrections and changes in the manuscript. Here is our response: 1. Moderate English changes required. Response: Thank you very much. We have carefully revised the manuscript. Since they are too trivial, we show them in the revised paper as the traces. 2. Chapter 4 should be remained from "Some Discussion" to Discussion Response: Thank you very much. We have revised it as you suggested. 3. Examples of chiral pharmaceuticals and the implication of stereochemistry in therapy can be introduced in the Introduction section Response: Thank you very much. We have revised the manuscript as you suggested: Old manuscript (P1, Line31):] L-dopa is very effective in treating Parkinson's disease, while D-dopa can cause serious side effects. Therefore… Revised manuscript: L-dopa is very effective in treating Parkinson's disease, while D-dopa can cause serious side effects. And for thalidomide, the R-(+)-enantiomer is an effective sedative, while the S-(-)-enantiomer can cause fetal abnormalities. Therefore ….

Reviewer 2 Report
can be accepted now
Author Response
The authors thank the comments and suggestions made by the reviewer very much. According to these comments and suggestions, we made corresponding corrections and changes in the manuscript. Here is our response:
- English language and style are fine/minor spell check required.
Response: Thank you very much. We have carefully revised the manuscript. Since they are too trivial, we did not list them here. They can be found in the revised paper as the traces.
